# Leaf Removal Impacted Jasmonic Acid Metabolism and AsA-GSH in the Roots of *Malus baccata* (L.) Borkh. under Suboptimal Low Root-Zone Temperatures

**Ping Dai [1], Meiling Zhai [1], Lijie Li [1], Huan Yang [1], Huaiyu Ma [1,2,]\* and Deguo Lyu [1,2,]\***

[1]  College of Horticulture, Shenyang Agricultural University, Shenyang 110866, China
[2]  Key Lab of Fruit Quality Development and Regulation of Liaoning Province, Shenyang 110866, China
\*  Correspondence: 2006500038@syau.edu.cn (H.M.); lvdeguo@syau.edu.cn (D.L.)

**Abstract:** In the early growing season in northern China, suboptimal low root-zone temperatures is a common abiotic stress that impairs root function and leaf development in fruit trees. In this study, we investigate the physiological role of leaves in jasmonate metabolism and the capacity of scavenging reactive oxygen species in *Malus baccata* (L.) Borkh. roots under suboptimal low root-zone temperatures. In the presence of intact leaves, suboptimal low root-zone temperatures significantly increased allene oxide synthase (AOS), jasmonate-resistant 1 (JAR), and jasmonic acid carboxyl methyltransferase (JMT) activities and transcription in jasmonate biosynthesis. Meanwhile, elevated endogenous jasmonic acid (JA), methyl jasmonate (MeJA), and jasmonate-isoleucine (JA-Ile) contents were also observed, as were significantly decreased glutathione reductase and dehydroascorbate reductase activities and AsA/DHA and GSH/GSSG ratios. Conversely, leaf removal substantially reduced AOS, JMT, and JAR activities and transcription at most time points and JA (6–24 h), MeJA (1–24 h), and JA-Ile (1–24 h) levels in roots, affecting key enzymes in the AsA–GSH cycle and the AsA/DHA and GSH/GSSG ratios in response to low-temperature treatment, as a result of a significant increase in malondialdehyde content. Thus, leaves are crucial for jasmonate metabolism in roots under suboptimal low root-zone temperatures, with leaf removal exacerbating root oxidative stress by altering JA signaling and AsA–GSH cycle activity.

**Keywords:** jasmonate; leaf removal; *Malus baccata* (L.) Borkh.; low-temperature stress; JA signal; oxidative stress

## 1. Introduction

A low temperature is a major abiotic stress that affects plant growth, development, and productivity [1–3]. Low-temperature stress leads to the accumulation of excessive reactive oxygen species (ROS), which causes protein denaturation and membrane lipid peroxidation and accelerates plant aging and death [4–9]. Studies have shown that low root-zone temperatures affects root and shoot function and growth [10]. Generally, suboptimal low root-zone temperatures is defined as a low temperature below the optimum temperature for plant growth and development but without causing fatal damage to the plant. Suboptimal low root-zone temperatures can significantly decrease plant height, stem diameter, root length, leaf area, and root and shoot biomass (dry weight). However, it increases the production of foliar hydrogen peroxide ($H_2O_2$), superoxide radical ($O_2^-$), and malondialdehyde (MDA) [11–13]. Hence, suboptimal low root-zone temperatures inhibit vegetative growth and causes oxidative stress in plants.

In plants, the ascorbate–glutathione (AsA–GSH) cycle is involved in the process of $H_2O_2$ elimination. The main enzymes in the AsA–GSH cycle include ascorbate peroxidase (APX), monodehydroascorbate reductase (MDHAR), dehydroascorbate reductase (DHAR), and glutathione reductase (GR). The substrates of AsA–GSH include ascorbate (AsA), dehydroascorbic acid (DHA), oxidized glutathione (GSSG), and reduced glutathione

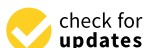



(GSH) [14–18]. Liu et al. [19] demonstrated that low-temperature stress substantially enhanced the activities of MDHAR, DHAR, APX, along with the AsA/DHA ratio, and greatly decreased the ratio of GSH/GSSG and GR activity in tomato. Li et al. [20] found that low-temperature stress (4 °C) significantly increased APX and GR transcription, as well as GSH levels in tea (*Camellia sinensis* (L.) Kuntze). In cucumber seedlings, low-temperature treatment significantly increased APX, DHAR, GR, and MDHAR activities and transcription, and substantially decreased AsA/DHA and GSH/GSSG ratios [21]. Our previous study showed that the AsA–GSH cycle in the roots of *Malus baccata* responded strongly to suboptimal low root-zone temperatures. That is, the APX activity level increased significantly, whereas GR activity significantly decreased [22]. Those findings suggest that the enzymes in the AsA–GSH cycle exhibit a different response to low temperatures.

Jasmonates (JAs), which include jasmonic acid (JA), methyl jasmonate (MeJA), and jasmonate isoleucine (JA-Ile) conjugate, are a part of hormone signal transduction pathway in plants and play important roles in response to abiotic stress [23,24]. Adverse conditions can affect endogenous JA levels [25,26]. Moreover, exogenous JA application enhances plant adaptation to abiotic stress by regulating endogenous JA levels [22,27,28]. Furthermore, the transcription levels of genes in the MeJA biosynthesis pathway (*SlAOS*, *SlOPR3*, and *SlJMT*) were significantly higher after injury in transgenic CfJMT-overexpressing tomato than in wild-type tomato [29]. Thus, CfJMT may participate in the abiotic stress response. MeJA application then maintained fruit quality during cold storage by enhancing APX and GR activities and increasing AsA and GSH levels [30–32]. Previous reports have indicated that low-temperature and salt stress significantly increase endogenous JA levels and improve abiotic stress resistance in *Artemisia annua* and tomato [25,33]. Our previous findings showed that suboptimal low root-zone temperatures variously affect allene oxide synthase (AOS), jasmonic acid carboxyl methyltransferase (JMT), and jasmonate-resistant 1 (JAR) involved in JA biosynthesis and AsA–GSH cycle activity in the roots of *M. baccata*. Furthermore, exogenous JA application could alleviate the oxidative stress induced by suboptimal low root-zone temperatures through enhancing the activities of antioxidant enzymes in the roots of *M. baccata*, such as APX and GR [22,34].

*M. baccata* is a widely used rootstock in the cool-climate apple-producing regions of northern China because of its strong environmental adaptation [35,36]. However, when the aboveground parts of the apple tree enter the growth stage in the early growth season, the roots often remain under about 5 °C, which impedes root function and nutrient translocation to the shoot. Zhang and Baldwin [37] showed that long-distance JA transport occurs between the leaves and roots by feeding mature leaves with [14]C-JA. Tracer experiments on plants under stress demonstrate that JA can be transported between roots and leaves, and the rate of downward transport is greater than that of upward transport [38,39]. Thus, it is believed that JA coordinates stress responses between the aboveground and underground parts of the plant [40,41]. However, little is known about the role of leaves in the adaptation of roots to suboptimal low root-zone temperatures. Therefore, to clarify the physiological function of leaves in the JA metabolism and antioxidant capability of apple roots exposed to suboptimal low root-zone temperatures, we performed leaf removal experiments in this study. A time course was utilized to examine endogenous JA levels, the activities and transcription levels of key enzymes in JA biosynthesis, the activities of the AsA–GSH cycle, and oxidative damage in the roots of *M. baccata* under suboptimal low root-zone temperatures. The results obtained in our work will provide a theoretical reference for clarifying the adaptation mechanism of apple roots to low temperature stress in cool-climate regions.

## 2. Materials and Methods

### 2.1. Plant Materials and Experimental Design

Seeds of *M. baccata* were purchased from Xiongyue town, Liaoning province, China. After being stored at 0–4 °C for 40 days, seeds were sown in a 50-well hole tray, with 1 seed for each hole for a total of 5 trays. The plug seedlings with 7–8 leaves were transplanted in plastic pots (12 × 13 cm) filled with sandy loam soil and cultivated in a greenhouse under

natural ambient light and temperature conditions (day/night temperatures, 26/18 °C; relative humidity, 50–60%). Pest- and pathogen-free seedlings with 15 leaves were selected for the subsequent experiments. The trial was conducted in an artificial climate room with day/night temperatures of 20 °C/10 °C and a 14 h/10 h light–dark photoperiod.

The experiment involved three treatments. For the control (CK), plants were kept in the artificial climate room without soil cooling. For the suboptimal low root-zone temperatures treatment (LF), the roots of the plants with leaves were maintained at 5 ± 0.5 °C in the artificial climate room. For the leaf-removal treatment (LR), all leaves of the plants were removed 6 h before the low-temperature treatment, and the roots of the plants without leaves were then exposed to suboptimal low root-zone temperatures (5 ± 0.5 °C) in the artificial climate room. During the whole experiment, the overground part in the three treatments were maintained with day/night temperatures of 20 °C/10 °C. The fine roots (1–3 mm) of 15 seedlings per treatment were collected at 0 h, 1 h, 6 h, 12 h, and 24 h, rinsed three times, frozen in liquid nitrogen, and stored at −80 °C prior to subsequent analyses. Each treatment comprised three biological replicates and each replicate comprised five seedlings.

### 2.2. Superoxide Radical ($O_2^-$), Hydrogen Peroxide ($H_2O_2$), and Malondialdehyde (MDA) Analyses

MDA was measured according to the thiobarbituric acid method reported by Velikova et al. [42]. Briefly, tissue samples (0.1 g) were homogenized in 3 mL of 10% trichloroacetic acid (TCA); the mixture was incubated at 4 °C for 30 min with rotation at 200 rpm. After centrifugation at $6000\times g$ for 10 min at 4 °C, the supernatant was collected and 2 mL of 0.6% TBA was added before placing in a boiling water bath for 15 min. After cooling, samples were centrifuged at room temperature at $12,000\times g$ for 10 min; finally, absorbances at 450, 532, and 600 nm were recorded.

The $O_2^-$ content was evaluated according to the Zhang et al. [43] method. Briefly, root tissues (0.1 g) were homogenized in 2 mL of 50 mM phosphate-buffered (pH 7.8); the mixture was incubated at 4 °C for 30 min with rotation at 200 rpm. After centrifugation at $6000\times g$ for 10 min at room temperature, 0.9 mL phosphate-buffered saline (PH 7.8) and 0.1 mL (10 mM) hydroxylamine hydrochloride were added to the supernatant, and the mixture was kept at 25 °C to react for 20 min, and then 1 mL of 17 mM p-amino benzenesulfonic acid and 1 mL of 7 mM α-naphthylamine were added to the mixture for incubation at 25 °C for 20 min before reading absorbance at 530 nm.

The $H_2O_2$ content was determined according to the method published by He et al. [44]. Briefly, root samples (0.1 g) were homogenized in 2 mL of 5% TCA. After centrifugation at 10,000 rpm for 10 min at 4 °C, 0.1 mL of 20% $TiCl_4$ and 0.2 mL ammonia was added to the supernatant, and the mixture was centrifuged at room temperature at 5000 rpm for 10 min. Then, 3 mL of 1 M $H_2SO_4$ was added to the precipitate to dissolve it before reading absorbance at 410 nm.

### 2.3. Antioxidant Enzyme Activity

Ascorbate (AsA), dehydroascorbic acid (DHA), reduced glutathione (GSH), and oxidized glutathione (GSSG) were determined according to the methods reported by Chen et al. [45]. Soluble proteins in the root samples were evaluated according to the Luo et al. [46]. Root tissue (0.1 g) was homogenized in 2 mL of cold phosphate-buffered saline (PBS; 0.1 M; pH 7.8) containing 0.5% (*v/v*) Triton X-100 and 100 mg polyvinylpolypyrrolidone. The mixture was shaken at 4 °C for 15 min, stored at 4 °C overnight, and centrifuged at $6000\times g$ and 4 °C for 30 min. The supernatants were then used to determine enzyme activity. Ascorbate peroxidase (APX) activity was determined according to the Polle et al. [47] method. Briefly, 1 mL of the reaction system was prepared with 50 μL supernatant, 650 μL $H_2O$, 100 μL KPP (pH 7.0), 100 μL AsA, and 100 μL $H_2O_2$. The change in absorbance decrease was monitored within 5 min, when ascorbate was oxidized at 290 nm with an extinction coefficient of 2.8 mM$^{-1}$ cm$^{-1}$. Dehydroascorbate reductase (DHAR) activity was measured according to the Chew et al. [48] method. Briefly, 1 mL of the reaction system was

prepared containing 30 μL supernatant, 100 μL $H_2O$, 570 μL KPP (pH 7.0), 100 μL EDTA, 100 μL GSH, and 100 μL DHA. The increase in absorbance within 3 min at 265 nm was then measured with an extinction coefficient of 14 $mM^{-1}$ $cm^{-1}$. Glutathione reductase (GR) activity was evaluated according to the Edwards et al. [49] method. Briefly, 1 mL of reaction system was prepared containing 50 μL supernatant, 550 μL $H_2O$, 100 μL EDTA, 100 μL GSSG, and 100 μL NADPH. The decrease in absorbance at 340 nm as NADPH was oxidized and was monitored with an extinction coefficient of 6.22 $mM^{-1}$ $cm^{-1}$. Monodehydroascorbate reductase (MDHAR) activity was measured by enzyme-linked immunosorbent assay kits (ELISA) (Shanghai Meilian Industrial Co. Ltd., Shanghai, China).

Reaction systems for the determination of superoxide dismutase (SOD) activity contained 50 μL supernatant, 1.75 mL KPP (pH 7.8), 300 μL NBT, 300 μL EDTA, 300 μL riboflavin, and 300 μL methionine. The reaction was started at a light intensity of 4000 lx for 20 min, and then absorbance was monitored at 560 nm. Each 50% inhibition of NBT reduction was used to define one unit (U) of SOD activity. The reaction system for the determination of peroxidase (POD) contained 200 μL supernatant, 3 mL mixed solution with guaiacol and hydrogen peroxide, and 800 μL KPP (pH 6.0). The increase in absorbance at 470 nm was monitored using an extinction coefficient of 26.6 $mM^{-1}$ $cm^{-1}$. Finally, the reaction system for the determination of catalase (CAT) contained 100 μL supernatant, 1 mL Tris-HCl (pH 7.0), and 1.7 mL $H_2O$. Absorbance at 240 nm was determined by adding 200 μL $H_2O_2$. A unit (U) of CAT was defined as a 0.1 decrease in absorbance per minute [49].

### 2.4. Jasmonic Acid (JA), Methyl Jasmonate (MeJA), and Jasmonate Isoleucine (JA-Ile) Levels

Endogenous JA was determined by ultra-performance liquid chromatography-tandem mass spectrometry (UPLC-MC/MC) according to the Engelberth et al. [50] method with certain modifications. Frozen root tissue (100 mg) was homogenized in 1 mL of acetone:citric acid (7:3, $v/v$) extractant mixture and shaken at 4 °C in the dark for 3 h. The centrifuge tube was uncovered and placed in an airstream under a fume hood overnight to evaporate the acetone solvent. Ether (700 μL) was added to the residue, the suspension was centrifuged at $10,000 \times g$ and 4 °C for 5 min, and the supernatant was collected. The residue was extracted once again as previously described, and then all supernatants were pooled and condensed with a vacuum concentrator (CV200; Beijing jiaimu Technology Co., Ltd., Beijing, China). The concentrate was then redissolved in 1 mL methanol and passed through a 0.22-μm (diameter, 13 mm) organic nylon filter. Then, 5 μL of the filtrate was injected into a Waters ACQUITY BEH column (C18; 1.7 μm; 2.1 mm × 100 mm; Water Corp., Milford, MA, USA) at 35 °C. The JA tuning parameters and conditions were optimized by injecting 500 ng $mL^{-1}$ of pure standard JA solution into the mass spectrometer. The UPLC system was coupled to a Waters Micromass Quattro Premier tandem quadrupole mass spectrometer (Waters Corp). The sample chamber temperature was 8 °C. Gradient elution was performed with water containing 0.1% ($v/v$) formic acid and methanol as the mobile phases. The eluent flow rate was 0.3 mL $min^{-1}$ and the run time was 5 min.

Endogenous MeJA and JA-Ile levels were evaluated by UPLC-MC/MC in multiple reaction monitoring mode (MRM) according to the Pan et al. [51] method with certain modifications. The modifications as described below: frozen root tissue (0.9 g) was homogenized in 500 μL of 2-propanol:$H_2O$:HCl (2:1:0.02, $v/v/v$) extraction buffer. The mixture was incubated on ice for 30 min, 1 mL dichloromethane was added, and the mixture was incubated at 4 °C for 30 min. The mixture was then centrifuged at $13,000 \times g$ and 4 °C for 5 min. The lower organic phase was collected into a 2 mL tube and evaporated under a nitrogen stream. The samples were redissolved in 400 μL of methanol containing 0.1% ($v/v$) formic acid. The sample mixture was passed through a 0.22-μm (diameter, 13 mm) organic nylon filter, and 10 μL of solution was injected into a Waters ACQUITY BEH column (C18; 1.7 μm; 2.1 mm × 100 mm; Waters Corp.) for UPLC-ESI-MS/MS analysis. Gradient elution was performed with water containing 0.1% ($v/v$) formic acid and methanol containing 0.1% ($v/v$) formic acid as the mobile phases. A flow rate of 0.3 mL $min^{-1}$ was used in all

analyses and the run time was 8 min. The MRM conditions for each compound are shown in Table 1.

**Table 1.** Mass spectrometry optimization conditions in multiple reaction monitoring mode.

| Compound | Scan Mode | Precursor Ion (m z$^{-1}$) | Product Ion (m z$^{-1}$) | Cone Voltage (V) | Collision Energy (eV) |
|---|---|---|---|---|---|
| JA | − | 209.1 | 59.0 | 32.0 | 14.0 |
| MeJA | + | 225.1 | 155.1 | 35.0 | 12.0 |
| JA-Ile | − | 322.2 | 130.2 | 45.0 | 18.0 |

## 2.5. Allene Oxide Synthase (AOS), Jasmonic Acid Carboxyl Methyltransferase (JMT), and Jasmonate-Resistant 1 (JAR) Activities

Root AOS, JAR, and JMT activities were measured by ELISA according to the He et al. [44] and Axelrod et al. [52] methods, with certain modifications. Frozen root powder (100 mg) was dissolved in 1 mL PBS (0.01 M; pH 7.4) and the solution was centrifuged at $4000 \times g$ and 4 °C for 15 min. The supernatant was used for subsequent enzyme activity determination. Enzyme solutions were extracted and analyzed with plant AOS, JAR, and JMT ELISA kits (Shanghai Meilian Industrial Co., Ltd., Shanghai, China) according to the manufacturer's instructions.

## 2.6. Total RNA Extraction and Gene Transcript Measurement

Gene transcription levels were determined as described by Li et al. [22]. Total root RNA was extracted and purified with a plant RNA extraction kit (No. R6827; Omega Bio-Tek, Norcross, GA, USA). RNA concentration and quality were determined by spectrophotometry (NanoDrop 2000; Thermo Fisher Scientific, Waltham, MA, USA) and agarose gel electrophoresis, respectively. The RNA volume required for reverse transcription was calculated according to the RNA concentration. One microgram of total RNA was used to synthesize first-strand cDNA in a PrimeScript RT-PCR Kit (No. DRR037A; Takara, Dalian, China) and stored at −20 °C until later use as a PCR reaction template. The PCR primers were diluted ten-fold and qRT-PCR was performed on each gene using 10 μL of 2 × SYBR Green Premix Ex Taq II (No. DRR820A; Takara, Dalian, China), 1 μL of cDNA, 0.8 μL of each of the positive and negative primers, and 2.4 μL double-distilled H$_2$O in a CFX384 Real Time System (No. CFX384; Bio-Rad Laboratories, Hercules, CA, USA). β-Actin was the internal reference gene (Table 2). Relative mRNA expression was calculated according to the $2^{-\Delta\Delta Ct}$ method [53]. Primer 5.0 (http://primer5.ut.ee/, accessed on 26 October 2019) was used to design the PCR primers.

**Table 2.** Primer sequences and annealing temperatures of allene oxide synthase (AOS), jasmonic acid carboxyl methyltransferase (JMT), and jasmonate-resistant 1 (JAR) in apple roots.

| Gene and Accession No. | Annealing Temperatures (°C) | Primer Sequences (5′–3′) |
|---|---|---|
| *AOS* (XM_008379143.2) | 56 | F: CCCTCCTCCTCTTCTGTTTCA R: CCGTTGACTGGTATTTCTGGA |
| *JMT* NM_101820.4 | 58 | F: AACTGAAGGAAGAAAAAGGTG R: TTGAGAGAGCCAATGAAGACT |
| *JAR1* HF10212-RA | 58 | F: GTGCCGACTTTTTCCTACTTT R: CCACTTCCACCACATCTCCTA |
| *β-Actin* | − | F: TGGTGAGGCTCTATTCCAAC R: TGGCATATACTCTGGAGGCT |

## 2.7. Statistical Analysis

All data were processed in Statgraphics (STN, St. Louis, MO, USA) and plotted with Origin 2019b software (OriginLab, Northampton, MA, USA). All values are the means ± standard error (SE). One-way ANOVA was used to identify significant differences

in mean values among the different treatments. Differences between means were considered significant at $p < 0.05$.

## 3. Results

### 3.1. Endogenous Jasmonates (JAs) Contents

In the present study, we measured the endogenous JAs contents in roots treated with suboptimal low root-zone temperatures (LF, with leaves) and leaf removal (LR, without leaves). Compared with the CK (control), the endogenous JA content in both the LF and LR treatments increased at all time points. In contrast, the JA content in the LR group was significantly higher than that in the LF treatment only at 1 h, before becoming substantially lower from 6 h to 24 h (Figure 1a).

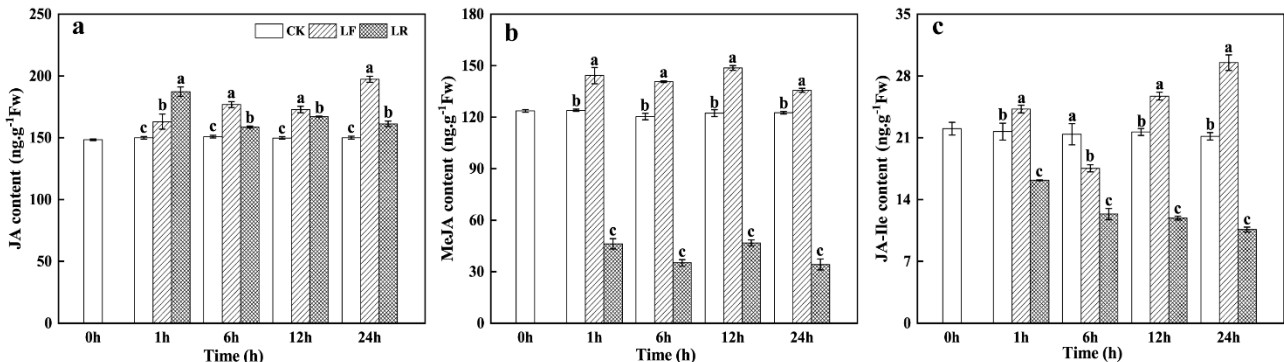

**Figure 1.** Effect of leaf removal on levels of endogenous jasmonic acid (JA) (**a**), methyl jasmonate (MeJA) (**b**), and jasmonate isoleucine (JA–Ile) (**c**) in the roots of *M. baccata* under suboptimal low root–zone temperatures. Means (±SE) of three replicates are shown for each treatment at each time point independently. Different lowercase letters on the bars indicate significant differences between treatments. CK: control; LF: suboptimal low root-zone temperatures treatment (with leaves); LR: leaf removal treatment under suboptimal low root-zone temperatures. The same abbreviations are used below.

Similarly, the endogenous MeJA content in the LF treatment remained significantly higher than that in CK from 1 h to 24 h. In contrast, the endogenous MeJA content in the LR treatment was significantly lower than that in the LF and CK treatments during 24 h of low-temperature treatment (Figure 1b).

Compared with the CK, the endogenous JA-Ile content was significantly higher in the LF treatment at 1 h, 12 h, and 24 h, but significantly lower at 6 h. In the LR treatment, however, the JA-Ile content dropped significantly from 1 h to 24 h and was significantly lower than that in both the CK and LF treatments, accounting for 36.0% that of LF treatment at 24 h (Figure 1c). Thus, leaf removal negatively regulated the accumulation of endogenous JAs in the roots of *M. baccata* under suboptimal low root-zone temperatures.

### 3.2. Activities and Transcription Levels of Key Enzymes in the Jasmonate Biosynthesis Pathway

Suboptimal low root-zone temperatures altered AOS, JMT, and JAR activities and expression levels in the roots of plants, both with and without leaves. That is, roots exposed to low-temperature stress had higher AOS, JMT, and JAR activities and exhibited various trends in their expression levels.

Compared with the CK, AOS activity in the LF treatment group significantly increased throughout the low-temperature treatment, with increases of 23.4–63.5%. In contrast, LR treatment substantially decreased AOS activity at 1 h, 12 h, and 24 h (15.3%, 33.3%, and 23.5% that of the LF treatment, respectively). However, AOS activity in the LR treatment was significantly higher than that in the CK at all the time points (Figure 2a). Moreover, compared with the CK, *AOS* was significantly downregulated in the LF treatment at 1 h but was significantly upregulated from 6 h to 24 h. In the LR treatment, *AOS* expression at

1 h was comparable to that in the LF treatment but was significantly higher than that in the LF treatment at 6 h and 24 h (1.87-fold and 4.12-fold, respectively), then substantially lower than that in the LF treatment at 12 h (Figure 2b).

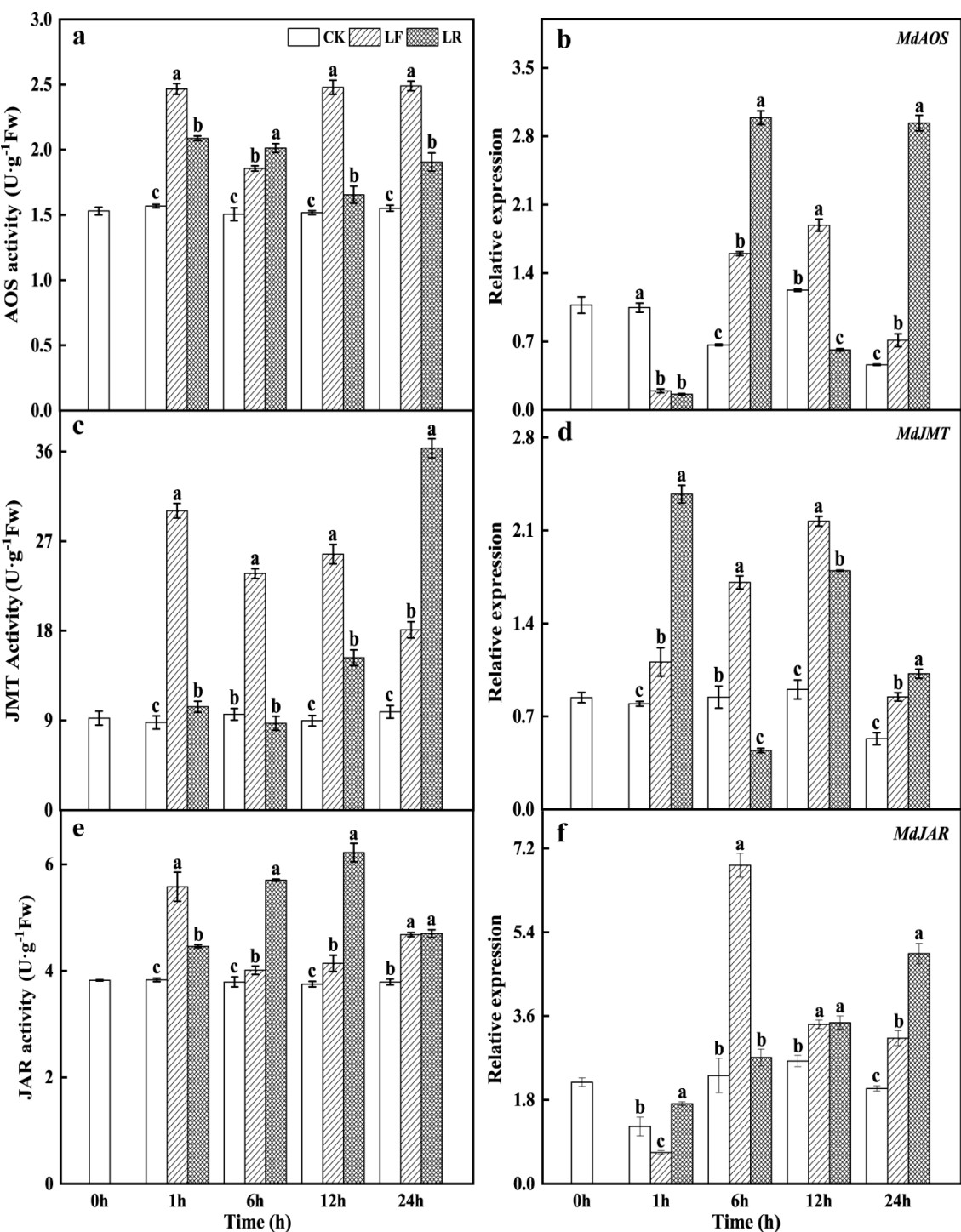

**Figure 2.** Effects of leaf removal on allene oxide synthase (AOS) activity (**a**), *AOS* expression (**b**), jasmonic acid carboxyl methyltransferase (JMT) activity (**c**), JMT expression (**d**), jasmonate–resistant 1 (JAR) activity (**e**), and JAR expression (**f**) in the roots of *M. baccata* under suboptimal low root–zone temperatures. Means (±SE) of three replicates per treatment at each time point independently. Different lowercase letters on the bars indicate significant differences between treatments.

JMT activity was significantly induced during 24 h of low-temperature treatment, peaking at 1 h (accounting for 2.41-fold that of the CK). JMT activity in the LR treatment was significantly lower than in the LF treatment from 1 h to 12 h, but increased significantly at 24 h, to 2.01 times higher than that in LF (Figure 2c). Additionally, *JMT* was significantly upregulated during 24 h of low-temperature treatment, peaking at 12 h (accounting for 2.4-fold that of the CK). LR treatment altered the trend of *JMT* expression, which was further upregulated at 1 h and 24 h, with a significant increase of 2.14 and 1.2 folds, respectively, in comparison with the LF treatment. Moreover, LR treatment downregulated *JMT* to significantly lower than that in both the LF and CK treatments (Figure 2d).

In both the LF and LR treatments, JAR activity was significantly higher than that in the CK. In the LF treatment, JAR activity increased significantly from 1 h to 24 h, and peaked at 1 h (accounting for 45.7% of the CK). In contrast to LF treatment, LR treatment significantly decreased JAR activity at 1 h (accounting for 20.1% that of the LF treatment), then strongly enhanced JAR activity at 6 h and 12 h (with a peak), with a significant increase of 42.1% and 50.2%, respectively (Figure 2e). With respect to *JAR* expression, compared with CK, low-temperature treatment significantly reduced *JAR* expression at 1 h. *JAR* expression in the LF treatment was significantly upregulated from 6 h to 24 h and peaked at 6 h (accounting for 2.94 fold that of the CK). LR treatment significantly upregulated *JAR* compared to that of CK during 24 h of low-temperature treatment. In contrast to the LF treatment, *JAR* expression in the LR treatment was evidently higher at 1 h and 24 h (2.56 times and 1.6 times higher, respectively), but significantly lower at 6 h (60.4% of in the LF) (Figure 2f). These results indicate that the three key enzymes that have a role in the JA synthesis strongly respond to suboptimal low root-zone temperatures stress, and that plays a crucial role in JA biosynthesis in the roots.

### 3.3. Superoxide Radical ($O_2^-$), Hydrogen Peroxide ($H_2O_2$), and Malondialdehyde (MDA) Contents

The $O_2^-$ content of the LF treatment was significantly higher than that of the CK throughout low-temperature treatment and was further increased in the LR treatment (Figure 3a). The $H_2O_2$ content of the LF treatment was also substantially higher than that of the CK, with a significant increase observed at 1 h, 6 h, and 24 h. In the LR treatment, the $H_2O_2$ content was lower than that in the LF treatment at 1 h and 6 h, with a significant decrease at 6 h, but significantly higher at 12 h and 24 h (Figure 3b).

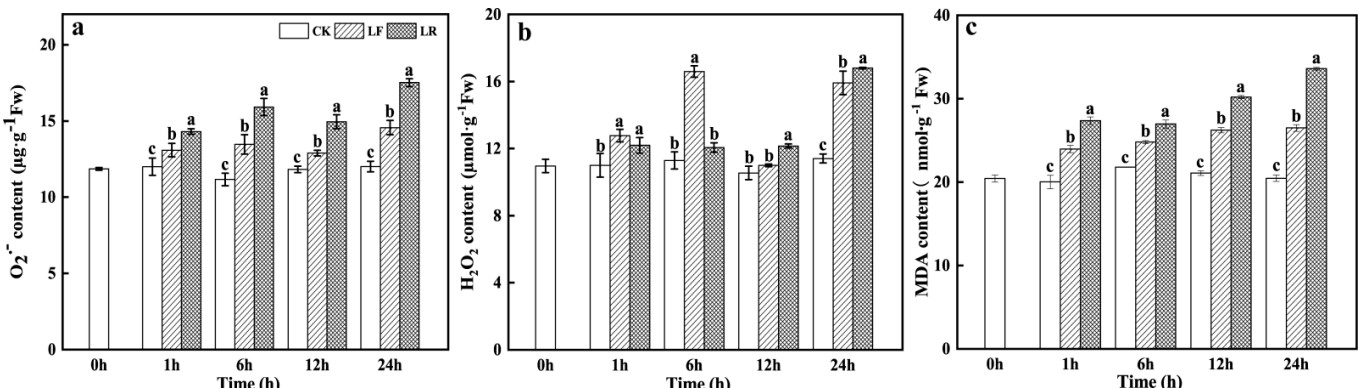

**Figure 3.** Effect of leaf removal on $O_2^-$ (**a**), $H_2O_2$ (**b**), and MDA (**c**) contents in the roots of *M. baccata* under suboptimal low root–zone temperatures. Means (±SE) of three replicates per treatment at each time point independently. Different lowercase letters on the bars indicate significant differences between treatments.

Compared with that in the CK, the MDA level in the LF treatment increased significantly and continuously throughout the low-temperature treatment (1.29 times that of CK) at 24 h. LR treatment further induced a significant increase in the MDA level which was 26.8% and 64.2% higher than that in the LF and CK treatments at 24 h, respectively

(Figure 3c). The above results demonstrate that suboptimal low root-zone temperatures stress led to oxidative damage in the roots, which was aggravated by leaf removal.

### 3.4. Superoxide Dismutase (SOD), Peroxidase (POD), and Catalase (CAT) Activities

The suboptimal low root-zone temperatures significantly enhanced SOD activity at 1 h and 24 h and significantly inhibited SOD activity at 6 h and 12 h. SOD activity at 1 h was similar in the LF and LR treatments, then became significantly higher in the LR treatment at 6 h and significantly lower at 12 h and 24 h (Figure 4a). LF treatment significantly enhanced POD activity at 1 h and 6 h which was further enhanced by LR treatment. POD activity in the LF group was then significantly inhibited at 12 h and was even further inhibited by LR treatment. Subsequently, POD activity in both the LF and LR treatments was significantly lower than that in the CK at 24 h but was significantly higher in the LR group than in the LF group (Figure 4b).

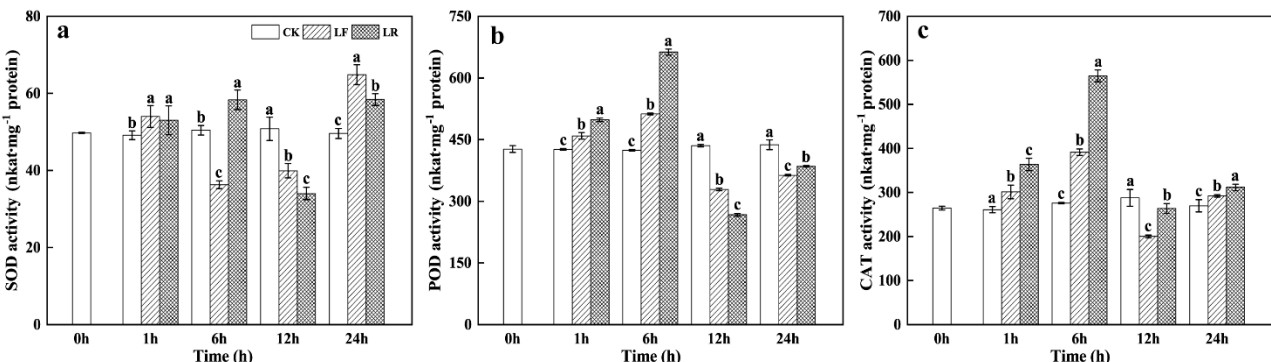

**Figure 4.** Effects of leaf removal on SOD (**a**), POD (**b**), and CAT (**c**) activities in the roots of *M. baccata* under suboptimal low root–zone temperatures. Data are means (±SE) for three replicates per treatment at each time point independently. Different letters on the bars indicate significant differences between treatments.

LF treatment significantly induced CAT activity at 1 h, 6 h, and 24 h, but significantly inhibited CAT activity at 12 h. In contrast, CAT activity was significantly higher in the LR treatment throughout the low-temperature treatment (Figure 4c).

### 3.5. Ascorbate–Glutathione (AsA–GSH) Cycle Activity

Compared with CK, both LF and LR treatments significantly inhibited APX activity at 1 h (Figure 5a). APX activity was then significantly enhanced by LF and LR treatments from 6 h to 24 h. Specifically, APX activity in the LF treatment peaked at 12 h and was significantly higher than that in the LR treatment. In contrast, APX activity in the LR group was significantly higher at 6 h and 24 h than that in the LF treatment (Figure 5a). Compared with CK, LF treatment significantly inhibited MDHAR activity at 1 h, which was further inhibited by LR treatment. Subsequently, MDHAR activities in the LF and LR treatments presented an increasing trend from 6 h to 24 h and were significantly higher than those in the CK. However, MDHAR activity in the LF treatment was significantly higher than that in the LR treatment (Figure 5b). The AsA/DHA ratio significantly decreased compared to that of CK by LF treatment at 1 h, 6 h, and 24 h, but significantly increased at 12 h. In contrast, the AsA/GSH ratio of the LR treatment was significantly lower than that of the LF treatment at 1 h and 12 h, but higher at 6 h and 24 h (Figure 5c).

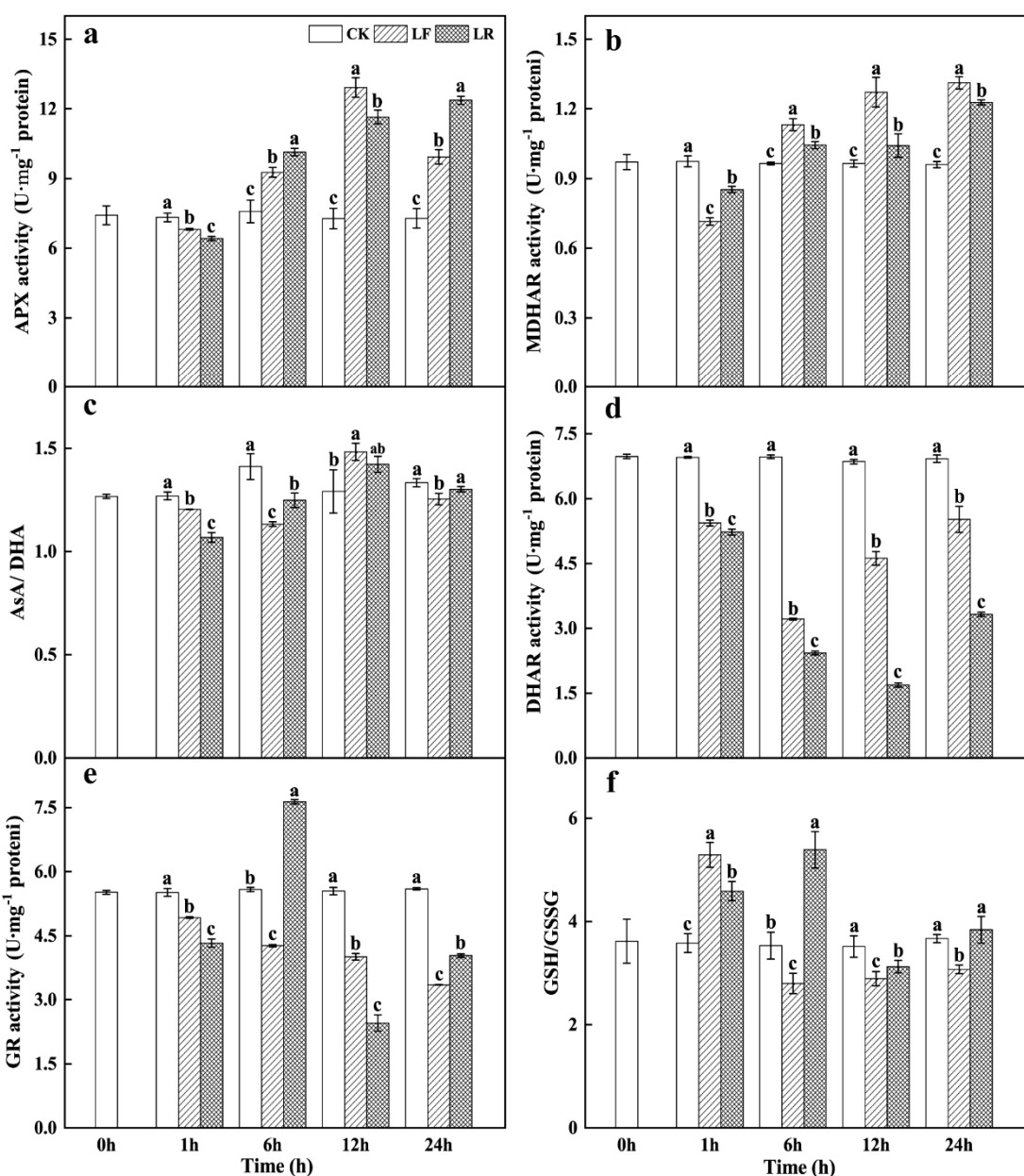

**Figure 5.** Effects of leaf removal on APX (**a**), MDHAR (**b**), AsA/DHA ratio (**c**), DHAR (**d**), GR (**e**), and GSH/GSSG ratio (**f**) in the roots of *M. baccata* under suboptimal low root–zone temperatures. Data are means (±SE) for three replicates per treatment at each time point independently. Different letters on the bars indicate significant differences between treatments.

Compared with the CK, LF treatment significantly decreased DHAR activity during 24 h of low-temperature treatment by 20.3–53.9%. DHAR activity was further inhibited by LR treatment at all time points, with a significant decrease from that in the LF group (Figure 5d). GR activity was significantly inhibited by LF treatment from 1 h to 24 h, with a decrease of 10.7% to 40.2% from that of CK. Compared with the LF treatment, GR activity in the LR treatment was significantly lower at 1 h and 12 h but was significantly higher at 6 h and 24 h, that is, 1.79 times and 1.20 times higher than that of the LF treatment, respectively (Figure 5e). Compared with the CK, the GSH/GSSG ratio in the LF treatment significantly increased at 1 h, and then significantly decreased from 6 h to 24 h. Conversely, LR treatment significantly increased the GSH/GSSG ratio from 6 h to 24 h, and the ratio peaked at 6 h to 1.93 times higher than that of the LF treatment. The GSH/GSSG ratio was significantly lower in the LR treatment than in the LF treatment only at 1 h (Figure 5f).

### 3.6. Principal Component Analysis

Principal component analyses were performed on the JA contents (JA, MeJA, and JA-Ile) and critical enzymes of the JA biosynthesis pathway (Figure 6a), JA contents and ROS levels, and key enzymes of the AsA–GSH cycle (Figure 6b) to evaluate the physiological effects of leaves under suboptimal low root-zone temperatures (LF: plants with leaves; LR: plants without leaves) (Figure 6). The LF and LR treatments were located in different principal component analysis (PCA) quadrants.

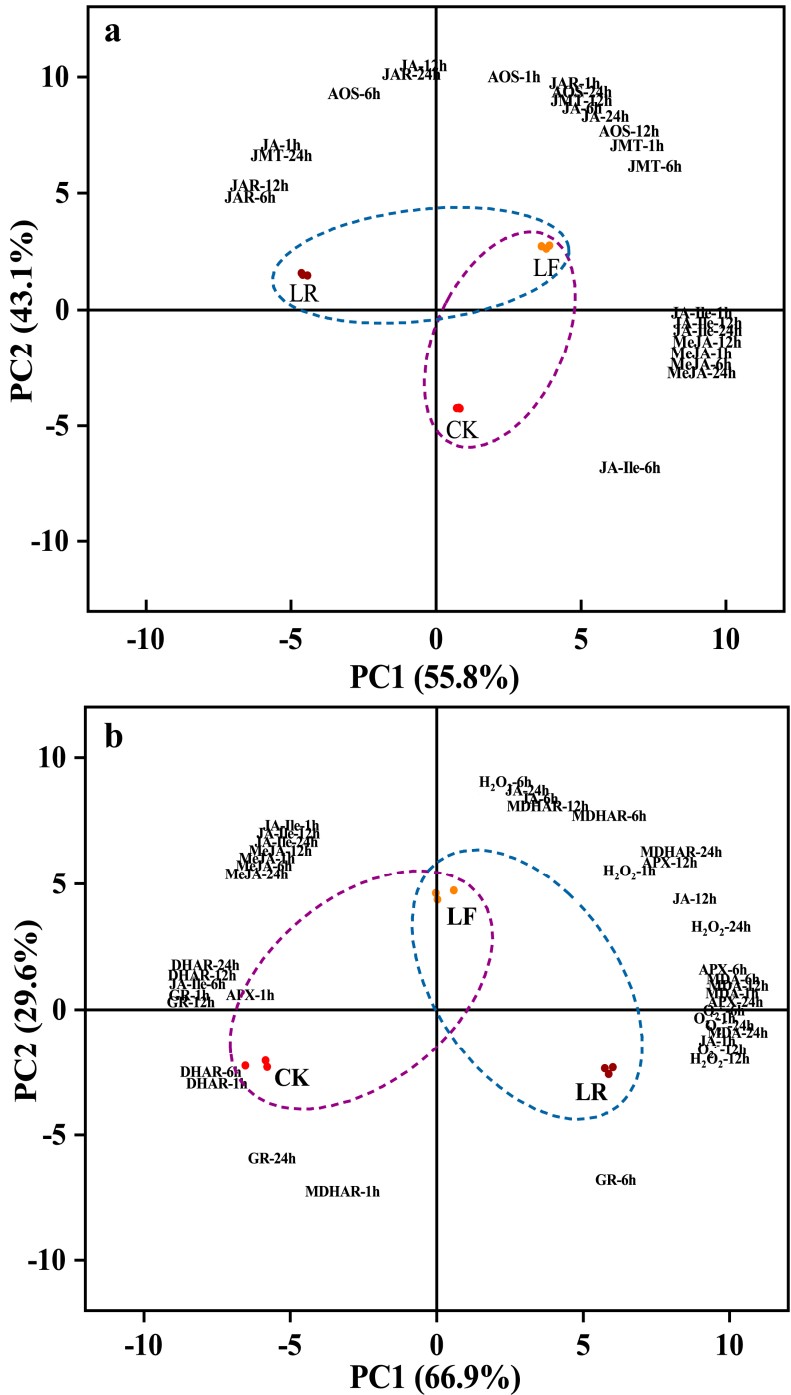

**Figure 6.** Plots showing the PCA of jasmonate levels, key enzymes of the jasmonate biosynthesis pathway (**a**), key enzymes of the ascorbate–glutathione cycle, and ROS levels (**b**) in the roots of plants with leaves (LF) and plants without leaves (LR) under suboptimal low root–zone temperatures.

Figure 6a shows that PC1 revealed the influence of suboptimal low root-zone temperatures on the JAs levels and the key enzymes in the JA synthesis, and explained 55.8% of the total observed variance. JA level (6 h, 24 h), JA-Ile level (1 h, 12–24 h), MeJA level (1–24 h), AOS activity (1 h, 12–24 h), JAR activity (1 h), and JMT activity (1–12 h) were the main positive contributors to PC1. PC2 revealed the effects of leaf removal on the JAs levels and the key enzymes in the JA synthesis under suboptimal low root-zone temperatures and explained 43.1% of the total observed variance. The main positive contributors to LR treatment were JA level (1 h), JAR activity (6–12 h), JMT activity (24 h), and AOS activity (6 h). In addition, the main negative contributors were JA-Ile level (1–24 h) and MeJA level (1–24 h).

Figure 6b shows that PC1 revealed the impacts of leaf removal on JAs levels, ROS levels, and the key enzymes in the AsA–GSH cycleunder suboptimal root-zone temperatures, and explained 66.9% of the total observed variance. The main positive contributors to LR treatment were MDA level (1–24 h), APX activity (6 h, 24 h), JA content (1 h), MDA content (1–24 h), $O_2^-$ content (1–24 h), and $H_2O_2$ content (12 h). PC2 revealed the effect of suboptimal low root-zone temperatures on JAs levels, ROS levels and the key enzymes in the AsA–GSH cycle and explained 29.6% of the total observed variance. JA content (6–24 h), MeJA content (1–24 h), JA-Ile (1, 12–24 h), APX (12 h), $H_2O_2$ content (1, 6, 24 h), and MDHAR activity (6–24 h) contributed positively to LF treatment.

## 4. Discussion

Low-temperature stress affects plant growth and development, as well as crop quality and yield [54]. Our preceding experiments demonstrated that a suboptimal low root-zone temperatures (5 °C) decreased plant height, root-to-shoot ratio, and leaf area index, and induced serious oxidative damage to the roots and leaves of *M. baccata* [22,55]. Subsequent application of exogenous MeJA alleviated negative impacts of suboptimal low root-zone temperatures on the aboveground plant parts by activating the JA signal system (unpublished data). JAs, as an essential long-distance signal in plants, can be transported between roots and leaves under stress [37–39,56,57]. Therefore, we speculate that leaves play an important role in plant adaptation to suboptimal low root-zone temperature stress.

The adaptation of plants to detrimental environments involves both spatial and functional partitioning of the shoots and roots. Signaling events are required to orchestrate whole-plant responses to environmental stimuli [58,59]. JA and its derivatives are key components in the signaling pathway that regulate the genes involved in plant defense against environmental stress [23,56,57,59]. MeJA, as a volatile JA derivative, acts as an intercellular signal and triggers signal production in the defense response [60]. JA-Ile mediates the initial binding of the jasmonate-ZIM domain (JAZ) protein to the SCF[COI1] complex, which degrades the JAZ complex and releases MYC2 [61,62]. Exposure of *Arabidopsis* to a low temperature (4 °C) for 1.5 h significantly increased its endogenous JA content and maintained it at a high level over 24 h [26]. Our previous study showed that suboptimal low root-zone temperatures significantly altered root and leaf JA levels. However, the JA level in the leaves peaked earlier than that in the roots [34]. Therefore, we speculated that the leaves might regulate the antioxidant capacity of the roots under suboptimal low root-zone temperatures via the JA signal.

In the present study, we examined JA metabolism and antioxidant capacity in the roots of *M. baccata* with and without leaves under suboptimal low root-zone temperatures for 1–24 h. The endogenous JA content of roots in the LR treatment (without leaves) was significantly higher than that in the LF treatment (with leaves) only at 1 h (1.1-fold) under suboptimal low root-zone temperatures and then became significantly lower from 6 h to 24 h (Figure 1a). Moreover, leaf-removal treatment significantly reduced endogenous MeJA and JA-Ile levels throughout the treatment (Figure 1b,c). The PCA results also showed that leaf removal had a negative impact on the levels of JA-Ile and MeJA (Figure 6a). Hence, leaf removal significantly negatively regulated JA, MeJA, and JA-Ile biosynthesis and, by extension, altered JA signal transduction in roots subjected to suboptimal low root-zone

temperatures. Therefore, we speculate that leaf removal impeded long-distance signal transduction, altering JA signals in roots under suboptimal low root-zone temperatures.

As key enzymes in the JA metabolic pathway, AOS, JAR, and JMT are involved in the regulation of plant growth and development, as well as adaptation to environmental stress [29,33,63]. AOS, JAR, and JMT activities and expression significantly impact endogenous JA levels [33,64]. Hu et al. [26] reported that *AOS* and *JAR1* were significantly upregulated in the leaves of *Arabidopsis* at 4 °C, thereby markedly increasing the endogenous JA levels in plant leaves, which demonstrates that the JA signal participates in the regulation of cold tolerance in *Arabidopsis*. AOS is the genuine starting enzyme in the jasmonate biosynthesis pathway [65]. Previous studies have shown that upregulation of JA biosynthesis genes such as *AOS* in *Camellia japonica* increases its endogenous JAs levels, thereby improving its cold stress tolerance [66]. Here, LF treatment (with leaves) significantly enhanced AOS activity and upregulated *AOS* from 6 h to 24 h. Conversely, LR treatment (without leaves) significantly decreased AOS activity relative to LF treatment, at all time points (accounting for 15.3–33.3% of the LF) except 6 h, and downregulated *AOS* at 12 h (with 3.1-fold that of the LF) (Figure 2a,b). This indicates that leaf removal significantly reduces endogenous JA levels in the roots by negatively regulating AOS. Secondary evidence from PCA data showed that AOS activity at 6 h was the main contributor to LR treatment (Figure 6a).

Jasmonate-resistant 1 (JAR1) is a terminal enzyme that conjugates JA to Ile [65,67–69], and JA-Ile initiates the JA signaling pathway [70,71]. *OsJAR1* upregulation induces JA-Ile accumulation and initiates JA signal transduction in rice [72]. Liu et al. [33] found that the expression level of *JAR1* and endogenous JA content in *A. annua* peaked at 6 h, indicating that JA signal was triggered by low-temperature stress. In the present study, suboptimal low root-zone temperatures significantly upregulated *JAR1* (6–24 h), enhanced JAR activity (1–24 h), and increased endogenous JA-Ile level (1 h, 12–24 h) in the roots of plants (with leaves, LF treatment), which implied the activation of JA signaling by low-temperature stress (Figures 1c and 2e). In contrast with the LF treatment, leaf removal (LR treatment) not only affected the JAR1 activity but also altered the JAR1 activity trend during 24 h of low-temperature stress. JAR1 activity in LR treatment significantly decreased at 1 h and peaked at 12 h (with 1.5-fold that of the LF treatment), whereas JAR1 activity in LF treatment peaked at 1 h (with 1.46-fold that of the CK) (Figure 2e). The PCA data showed that the contribution of JAR activity to the LR treatment was negative at 1 h (Figure 6a). The change in JAR1 activity would result in a different accumulation of JA-Ile between the LF treatment and the LR treatment. Subsequently, different JA-Ile levels would arouse JA signals with various physiological functions. That is, leaf removal altered the JA signal in the roots through changing JA-Ile levels.

JMT promotes MeJA biosynthesis by using JA as its substrate [73]. Previous studies have reported that *AtJMT* overexpression in transgenic plant lines significantly increases endogenous MeJA levels and upregulates JA-responsive genes related to plant immunity [74]. Thus, MeJA is considered a pivotal trigger of jasmonate-related defensive responses that acts as a signal carrier within and among plants [74–76]. In the present study, LF treatment significantly enhanced JMT activity and *JMT* transcription, as well as endogenous MeJA levels, in roots (Figures 1b and 2c,d). This suggests that JMT in *M. baccata* roots may positively respond to suboptimal low root-zone temperature stress and participate in long-distance signal transduction by promoting MeJA synthesis. In contrast, leaf removal (LR treatment) significantly inhibited JMT activity from 1 h to 12 h (with a decrease of 2.5-3.4-2.9 fold, respectively) (Figure 2c). Simultaneously, endogenous MeJA levels in LR treatment were significantly lower than those in LF treatment (Figure 1b). The PCA data confirmed that JMT activity had a negative relationship with the LR treatment at 1–12 h (Figure 6a). The above results indicate that leaf removal impairs the long-distance transmission of JA signals by lowering MeJA levels in the roots under suboptimal low root-zone temperatures.

Moreover, the activities of AOS, JMT, and JAR peaked at various time points, and the *LOX*, *AOS*, *JMT*, and *JAR* transcription levels also differed in the presence or absence

of leaves under suboptimal low root-zone temperatures (Figure 2). Li et al. [66] reached a similar conclusion after studying the response of the JA signaling pathway in *Camellia japonica* to cold acclimation at 4 °C. PCA analysis also proved that leaf removal had a genuine impact on JA synthesis in the roots of *M. baccata* under suboptimal low root-zone temperatures (Figure 6). These results indicate that the leaves strongly influence the response of JA biosynthesis and the JA signal to suboptimal low root-zone temperatures.

ROS-scavenging enzymes such as SOD, POD, and CAT are vital to the plant antioxidant system and protect cells from ROS damage [77]. In the present study, the aforementioned antioxidant enzymes responded differently to a suboptimal low root-zone temperatures (Figure 4). SOD activity at 1 h and 24 h was significantly higher in the LF treatment than in CK but was significantly inhibited at 12 h and 24 h by leaf removal (LR treatment) (Figure 4a). $O_2^-$ content in the LR treatment was significantly higher from 1 h to 24 h than that in the CK (Figure 3a). These results suggested that leaf-removal treatment weakened the ability of the roots to eliminate $O_2^-$ under low-temperature stress, resulting in excessive accumulation of $O_2^-$. $O_2^-$ is highly active and unstable and can react with Fe-S clusters in proteins or generate $H_2O_2$ catalyzed by SOD [78]. $H_2O_2$ content at 1 h and 6 h was significantly lower in the LR treatment than in the LF treatment, but was significantly higher from 12 h to 24 h (Figure 3b). Moreover, the activities of POD and CAT in the LR treatment were significantly enhanced at most time points during lthe ow-temperature treatment (Figure 4b,c). Based on the above results, we speculate that the significant decrease in $H_2O_2$ content at 1 h and 6 h was partly due to enhanced POD and CAT activities; however, the more important reason was the reduced conversion of $O_2^-$ to $H_2O_2$. From 12 h to 24 h, the significant increase of CAT (at 12 h and 24 h) and POD (at 24 h) could not effectively stabilize the $H_2O_2$ metabolic balance in the roots (Figures 3b and 4b,c) because the MDA levels in the LR treatment continued to increase from 1 h to 24 h (Figure 3c).

The AsA–GSH cycle is considered an effective ROS-scavenging mechanism in plants. APX, MDHAR, DHAR, and GR are the main components of the cycle, which scavenge excess $H_2O_2$ in plants and alleviate oxidative damage caused by low-temperature stress [8,79]. The AsA–GSH cycle maintains redox homeostasis by regulating the AsA/DHA and GSH/GSSG ratios [80,81]. Our previous study revealed that the enzyme activities and gene expressions of APX and GR activity were strongly responsive to suboptimal low root-zone temperatures and exogenous JAs [22,55]. Moreover, exogenous JAs can enhance the ability to eliminate ROS in the roots by increasing APX and GR activities under suboptimal low root-zone temperatures [22]. Therefore, we assumed that the activity of the AsA–GSH cycle was altered by the leaf-removal treatment, which altered JA signaling under suboptimal low root-zone temperatures. In the present study, the response characteristics of the four key enzymes in the AsA–GSH cycle varied under suboptimal low root-zone temperatures (Figure 5a,b,d,e). These results were consistent with the reports of Luo et al. [82] and Ramazan et al. [83]. Compared with the LF treatment, the LR treatment had various effects on the main components of the AsA–GSH cycle. LR treatment significantly increased APX and GR activities, as well as AsA/DHA and GSH/GSSG ratios, only at 6 h (accounting for 10.2% and 92.6% that of LF treatment, respectively) (Figure 5a,c,e,f). Combined with the results of $H_2O_2$ content at 6 h, this suggests that APX and GR might contribute to $H_2O_2$ elimination upon LR treatment (Figures 3b and 5a,e). At other time points, APX activity, GR activity, and AsA/DHA and GSH/GSSG ratios exhibited similar changing trends during LR treatment. However, MDHAR and DHAR activities in the LR treatment decreased significantly from 6 h to 24 h (Figure 5). From 12 h to 24 h, $H_2O_2$ content increased sharply and was significantly higher upon the LR treatment than upon the LF treatment (Figure 3b). On the one hand, this implies that MDHAR and DHAR are more sensitive to low temperatures and leaf removal in the AsA–GSH cycle. On the other hand, the significant decrease in MDHAR and DHAR might explain the weak ability of the AsA–GSH cycle to scavenge $H_2O_2$. The PCA data also demonstrated that neither DHAR nor MDHAR played positive roles in ROS scavenging under LR treatment (Figure 6b). Here, the MDA content in LF and LR treatments confirmed that low root-zone temperatures caused oxidative stress in

the roots of *M. baccata*, and leaf removal further aggravated the oxidative damage under low-temperature treatment (Figure 3c). Therefore, we suggest that leaf removal disrupts the equilibrium between ROS generation and quenching partly by altering the activities of antioxidant enzymes when plants are subjected to suboptimal low root-zone temperatures. In addition, the findings on apple seedlings revealed the details of the metabolism and molecular changes in jasmonate metabolism, which would be expected to have similar effects in developed plants.

## 5. Conclusions

In this study, a suboptimal low root-zone temperatures induced oxidative stress in the roots of *M. baccata*, and the resulting oxidative damage was further aggravated by leaf removal, as shown by significantly higher levels of MDA and ROS. The results of this study indicate that leaf removal can alter JA signaling by regulating endogenous JA, MeJA, and JA-Ile levels under suboptimal low root-zone temperatures. As our previous studies demonstrated that the AsA–GSH cycle is strongly responsive to exogenous JAs under suboptimal low root-zone temperatures, we hypothesize that leaf removal impacts antioxidant enzyme activities and disturbs the ROS metabolic balance by altering JA signals under suboptimal low root-zone temperatures. In other words, leaves are essential for the expression of *M. baccata* root adaptability to suboptimal low root-zone temperatures stress.

**Author Contributions:** Conceptualization, D.L. and H.M.; methodology, P.D. and H.Y.; software, P.D., M.Z. and L.L.; validation, D.L. and H.M.; formal analysis, P.D., M.Z. and L.L.; investigation, P.D.; resources, D.L. and H.M.; data curation, P.D. and H.Y.; writing—original draft preparation, P.D.; writing—review and editing, H.M.; visualization, H.Y.; supervision, D.L. and H.M.; project administration, H.M.; funding acquisition, H.M. and D.L. All authors have read and agreed to the published version of the manuscript.

**Funding:** This research was funded by the National Natural Science Foundation of Liaoning [Grant No. 2019-MS-269], and the China Agriculture Research System of MOF and MARA [Grant No. CARS-27].

**Institutional Review Board Statement:** Not applicable.

**Informed Consent Statement:** Not applicable.

**Data Availability Statement:** The datasets generated and analyzed during the current study are available from the corresponding author upon reasonable request.

**Conflicts of Interest:** The authors declare no conflict of interest.

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
