# Peer review of "Leaf Removal Impacted Jasmonic Acid Metabolism and AsA-GSH in the Roots of Malus baccata (L.) Borkh. under Suboptimal Low Root-Zone Temperatures"

_horticulturae, doi:10.3390/horticulturae8121205_

Round 1
Reviewer 1 Report
In my opinion the manuscript is well written and organized and the results and conclusions are sound. The figures are clear. Perhaps and extra figure summarizing in an image the main conclusions would be useful for readers to easily grasp what is happening with the different enzymes and compounds in relation to cold stress and the presence or absence of leaves.
Reviewer 2 Report
The manuscript concerns the metabolism of jasmonates in plant response to low temperature, being one of the major abiotic stresses for the cultivated plants. The chosen experimental model was apple tree, Malus baccata, a popular and important crop plant. The manuscript is interesting, valuable and well written. Methods were selected properly, the results are clearly presented and convincingly discussed.
Nevertheless, the experiments performed in this study was conducted on apple tree young seedling, and the problem of environmental stress concerns usually the fully developed plants. Certainly, it is understandable that such experiments cannot be conducted on fully developed apple trees in the controlled conditions. However, I would expect just a short comment on that issue in the discussion – a notice that the experiments were made on seedlings to revealed the details of the metabolic and molecular changes in jasmonate metabolism, and it can be expected that similar effects would occur in developed plants.
Line 98. Please insert the space between pest and “and” in the statement “Pest-and pathogen-free”
Please correct the references (there are some not necessarily underlined parts).
Reviewer 3 Report
Dear Authors
The current manuscript entitled “Leaf Removal Impacted Jasmonic Acid Metabolism and AsA-GSH in the Roots of Malus baccata Borkh. under Suboptimal Low Root-zone Temperature” investigates the physiological role of leaves in jasmonate metabolism and the capacity of scavenging reactive oxygen species in Malus baccata Borkh. The findings suggested that leaves are crucial for jasmonate metabolism in roots under suboptimal low root-zone temperature by altering JA signaling and AsA–GSH cycle. The abstract is well summarised and introduction is organised and informative, although the materials/method and results section need attention to include more information. Please find my suggestions here below for further improvement of the manuscript.
1. Line 102-111. The roots of the plants with leaves were maintained at 5 ± 0.5 °C”, does it mean that complete plant was maintained in artificial chamber under 5 ± 0.5 °C. If so, Please write it simple that the plants (with and without leaves) were maintained in defined conditions.
2. Line 112. “2.2. Superoxide Radical, Hydrogen Peroxide (H2O2), and Malondialdehyde (MDA)” Please describe the methods in detail. Only providing reference is not enough.
3. Line 117. Similarly the antioxidant enzyme activities should be explained in detail, especially the calculations involved in the method should be explained, which may be easier for readers to understand.
4. Line 175. How much sample was taken for RNA extraction?
5. Line 191. Table may include the accession number of corresponding genes.
6. Figure 1 legend does not include the fig b, please check.
7. How did you maintain the low temperature in root zone only?
8. Fig 2. At 0 hr, there is only CK?
9. Similarly fig 3,4 and 5 show only CK at 0hr?
Thank you
Regards
Reviewer 4 Report
The manuscript is scientifically sound, presented in a well-organized manner and in standard (but sometimes erroneous) English. There is a substantial problem with statistical analyses and significance letters above histogram bars may be faulty inserted. Some examples are labelled in the MS attached herewith. It is necessary to professionally re-assess statistical testing to make differences between the treatments properly presented. Moreover, modifications in both Results and Discussion sections have to be made accordingly. Regarding explanation of PCA results in L345-359, it should be resolved with a help of an experienced statistician, since the two plots are wrongly interpreted in the text. Therefore, the whole paragraph should be rewritten as well as parts of Discussion (L397-398, L420-421, L434-435, L450-451, L505-507). More specific comments related to this issue are provided in the file attached in the report. Furthermore, there are numerous constructive comments and notifications that the authors may find useful.

Reviewer 5 Report
The manuscript entitled "Leaf Removal Impacted Jasmonic Acid Metabolism and AsA-GSH in the Roots of Malus baccata Borkh. under Suboptimal Low Root-zone Temperature" by Dai et al., was reviewed for publication in Horticulturae (Horticulturae-2068726).
Comments
Figure legend 1, define CK, LF and LR in the legend.
General comment, I think in general for the results (Sections 3.1 - 3.5), you should start by describing the generalities comparing the CK to the LF and LR treatments, then go in to the details comparing the two treatments, LF and LR. See descriptions below.
Figure 1, what I see from the graphs is that the JA content increases at all time points in both LF and LR compared with CK, while JA is higher in LR at 1h, and lower than LF at other time points. In contrast, MeJA is higher only in LF compared with CK, while lower in LR at all time points. In general, the JA-Ile content follows a similar pattern as MeJA, except at the 6h time point, when content in both LF and LR is lower than CK. At all other time points, JA-Ile is higher in LF than CK, and lower in LR than both CK and LF. This is not clearly stated in the results and the description of the results could be improved.
Figure 2, lines 228-236. Again, I think it is more straight forward to say that the AOS activity in the LF and LR was higher than the CK at all time points. Then, compare the LF and LR. The AOS activity was higher in the LF compared to LR at all time points except 6h, when it was slightly but significantly higher.
Lines 305-308, indicate the Figure that corresponds to the results being described.
Line 372, I don’t understand why the data that show “MeJA alleviated the negative impacts of suboptimal 370 low root-zone temperature on the aboveground plant parts by activating the JA signal 371 system (unpublished data)”. To start the discussion out by citing data that is not derived from the current study, or a previous study, is not very acceptable for me.
372-376, it is stated that the main site of JA biosynthesis occurs in the leaves, then transport of JA from roots to leaves is an important to adapt to low root-zone temperature stress. I’m unclear on what is being proposed here, given that the two statements seem to contradict each other, and in the current study, it appears that JA, MeJA and Ile-JA content in the roots decrease more when the leaves are removed, which suggests these compounds are coming from the leaves.
Lines 387-388, again, pertinent data cited as unpublished, doesn’t seem appropriate.
Round 2
Reviewer 3 Report
Dear Authors
Thank you for answering all the queries, i do not have any further questions.
Regards
Author Response
Dear Editor and Reviewer:
Thank you very much again for your valuable advice and the reviewers’ constructive comments to our manuscript entitled ‘Leaf Removal Impacted Jasmonic Acid Metabolism and AsA-GSH in the Roots of Malus baccata (L.) Borkh. under Suboptimal Low Root-zone Temperature’, which are very helpful for improving our manuscript (horticulturae-2068726).
Reviewer 4 Report
The authors have meticulously addressed all the concerns raised by the reviewers. I deem the manuscript would be acceptable for publication in this form.
Author Response

(The authors gave the same response as above.)

Reviewer 5 Report
A revised version of the manuscript entitled "Leaf Removal Impacted Jasmonic Acid Metabolism and AsA-GSH in the Roots of Malus baccata Borkh. under Suboptimal Low Root-zone Temperature" by Dai et al., was reviewed for publication in Horticulturae (Horticulturae-2068726). The authors did a very good job to correct the manuscript, and I only have a few minor corrections to suggest below.
Comments
Line 97, delete “be”, to correct to “will provide”
Line 204, according to Pan et al. [51] with certain modifications. What are the certain modifications, those that are listed in the following sentences? If yes, then better to write with modifications as described below.
Line 517, “with a decrease of”, would be correct.
